# Tailoring vapor film beneath a Leidenfrost drop

An Li ⓘ[1,2,5], Huizeng Li ⓘ[1,5] ✉, Sijia Lyu ⓘ[3], Zhipeng Zhao ⓘ[1,2], Luanluan Xue[1,2], Zheng Li[1], Kaixuan Li[1,2], Mingzhu Li ⓘ[1], Chao Sun ⓘ[3,4] ✉ & Yanlin Song ⓘ[1,2] ✉

For a drop on a very hot solid surface, a vapor film will form beneath the drop, which has been discovered by Leidenfrost in 1756. The vapor escaping from the Leidenfrost film causes uncontrollable flows, and actuates the drop to move around. Recently, although numerous strategies have been used to regulate the Leidenfrost vapor, the understanding of surface chemistry for modulating the phase-change vapor dynamics remains incomplete. Here, we report how to rectify vapor by "cutting" the Leidenfrost film using chemically heterogeneous surfaces. We demonstrate that the segmented film cut by a Z-shaped pattern can spin a drop, since the superhydrophilic region directly contacts the drop and vaporizes the water, while a vapor film is formed on the superhydrophobic surrounding to jet vapor and reduce heat transfer. Furthermore, we reveal the general principle between the pattern symmetry design and the drop dynamics. This finding provides new insights into the Leidenfrost dynamics modulation, and opens a promising avenue for vapor-driven miniature devices.

Drops boiling, exploding, bouncing, and traveling on hot surfaces not only bring novel phenomena, but also play significant roles in spray cooling, fuel combustion, and microfluidics[1–22]. The temperature of the solid surface is one of the key factors in determining drop behavior. A unique transformation of the drop arises when increasing the surface to a critical temperature, known as the Leidenfrost point, where an intact vapor film detaches the drop from the supporting substrate[23,24]. In the Leidenfrost state, the slowed evaporation because of the poor heat conductivity of vapor, coupled with the eliminated adhesion resistance, enables a long-period actuation of the drop by the beneath vapor flow.

Recently, amounts of studies have attempted to modulate the Leidenfrost drop vapor by constructing elaborate topographical structures or designing specific surface wettability[10,25–31], since promoting the vapor film formation can repress explosively nucleate boiling[32] and realize drag reduction[33,34], while the inhibited vapor

accumulation will facilitate spray cooling[22] and improve combustion efficiency[35]. Although excellent works have partly achieved vapor guidance using physical structures, the rational design of chemical heterogeneity for efficient and precise drop vapor dynamics regulation is still in its infancy.

Here, we report a new discovery that the Leidenfrost film can be arbitrarily tailored to rectify the vapor dynamics by surface heterogeneity. As exampled by the drop gyro on a Z-shaped patterned wettability surface, the beneath "intact" Leidenfrost film is cut into two separate portions, which modifies the generated vapor into two unbalanced flows, resulting in a force couple to spin the drop. The underlying mechanism is analyzed and verified by establishing a reasonable physical model and adjusting the wettability pattern. More importantly, we propose a general principle to regulate the drop vapor dynamics by designing the symmetry of the wettability pattern, and

[1]Key Laboratory of Green Printing, CAS Research/Education Center for Excellence in Molecular Sciences, Beijing National Laboratory for Molecular Sciences, Institute of Chemistry, Chinese Academy of Sciences, 100190 Beijing, P. R. China. [2]University of Chinese Academy of Sciences, 100049 Beijing, P. R. China. [3]Center for Combustion Energy, Key Laboratory for Thermal Science and Power Engineering of Ministry of Education, Department of Energy and Power Engineering, Tsinghua University, 100084 Beijing, P. R. China. [4]Department of Engineering Mechanics, School of Aerospace Engineering, Tsinghua University, 100084 Beijing, P. R. China. [5]These authors contributed equally: An Li, Huizeng Li. ✉e-mail: lihz@iccas.ac.cn; chaosun@tsinghua.edu.cn; ylsong@iccas.ac.cn

further demonstrate the potential applications in micromechanical drive and directional drop transport.

## Results

### Drop gyro on hot patterned wettability substrate

The evolution of the drop sitting on a hot superhydrophobic surface is shown in Fig. 1a, b. The substrate is above the Leidenfrost temperature, and the constructed vapor film under the drop contributes a nearly "frictionless" move. By comparison, the drop placed on a hot chemically heterogeneous surface displays a distinct behavior (Supplementary Movie 1). The Z-shaped superhydrophilic pattern on the superhydrophobic surface consists of perpendicular lines with 100 μm in width and 1.5 mm, 3 mm, and 1.5 mm in length, respectively (Fig. 1c). The contact angle characterization of different regions is provided in Supplementary Fig. 1. As the snapshots of the synchronized oblique and side views displayed in Fig. 1d, spontaneous drop spin with a speed of 800 rpm occurs on the chemically heterogeneous substrates. Considering the drop with a radius of 2 mm and a rotational speed of 85 rad s$^{-1}$, the Weber number $We = \rho\omega^2 R^3/\sigma$ is of order 1, implying that the balance between centrifugal force and surface tension shapes the drop ellipsoidal.

We use two parameters to quantitatively describe the spinning drop, that is, the width of the drop projection $W$ based on the side view recording, and the drop centroid position $X$, as marked in Fig. 1d (the side view at $t = 0$ ms). The variations of these parameters as functions of time are plotted in Fig. 1e. The centroid position of the drop remains almost fixed during the spinning process (blue dots in Fig. 1e), while the drop projection width varies with time (red dots in Fig. 1e). The variation of the projection width can be well described by the function $W \sim \cos(0.17t - 0.1)$ (the black curve in Fig. 1e), which indicates that the drop can spin steadily during the investigated time frame, with a rotation period $T = 2\pi/0.17 = 74$ ms. In addition, we find that the drop spins on the condition that the substrate is heated to an appropriate temperature range. At a low temperature, the drop slightly shakes on the superhydrophilic pattern, while a vigorous detaching from the pattern happens at a high temperature (Supplementary Movie 2). Meanwhile, the volume of the drop will surely affect the drop behaviors. Figure 1f summarizes the substrate temperature and the drop volume required for the drop spin. The droplet spinning system operates in a limited temperature range from 115 to 135 °C, which is between the "cold Leidenfrost state"[19] and the "normal Leidenfrost state", that is, in a range where the hydrophilic region is wetted by the liquid, but not the hydrophobic one. Detailed discussions are provided in Supplementary Information.

### Mechanism of the drop gyro

To elucidate the origin of the spin, we analyze the drop dynamics and plot the evolution of the rotational speed $\omega$ versus time $t$ in Fig. 2b. After being placed on the hot substrate, the drop mainly experiences four evolution stages. In stage I (0–14 s), the drop shakes slightly in a spherical morphology, with no obvious spin observed ($\omega = 0$). Then the drop starts to spin and the rotational speed increases rapidly to 600 revolutions per minute (rpm) within 2 s (stage II), accompanied by the shape change from spherical to ellipsoidal. The steady spin lasts more than 70 s, with the rotational speed gradually growing to 1400 rpm (stage III). Meanwhile, the drop shrinks due to vaporization and the aspect ratio further enlarges. Finally, the drop spin ceases in stage IV as the drop size becomes smaller than the pattern (Supplementary Movie 3).

The pre-heating procedure (stage I) manifests that the rotating occurs when the drop raises to a certain temperature, which may trigger several effects accounting for the drop spin. The first possible effect is the Marangoni flow caused by the temperature difference inside the drop[36]. We utilize an infrared camera to record the heated drop, as shown in Supplementary Fig. 2 and Supplementary Movie 4. A temperature difference indeed exists between the top and bottom of the drop with a radius of 3 mm, while the small droplet with a 1 mm radius seems to manifest a uniform temperature distribution[18]. However, we find that the small droplet can also spin on the hot substrate (Supplementary Fig. 3 and Supplementary Movie 5), thus excluding the Marangoni effect. Second, the superhydrophilic pattern leads to asymmetric surface waves[37], caused by the bubble breakages, which may transport differentiated matter from the drop bottom to the top, making the drop spin possible. But the surface waves normally company with an obviously uneven drop surface. The quite smooth surfaces, as shown in Fig. 1d and Supplementary Movie 1, indicate that the surface wave effect could not cause the spin. The other effect of Rayleigh-Taylor instability[38] often occurs in centimeter-scale drops, which are much larger than our spinning drop.

Interestingly, a conspicuous "sizzle", which may be caused by bubbles bursting[39] or air rushing through a small gap[40], emerges when the drop starts to spin (Supplementary Fig. 4). The sound inspires us to examine the bottom of the drop, since the escape of the vapor between the drop and the substrate can generate a viscous force that may actuate the drop[41,42]. A high-speed camera provides the bottom view of the drop (Fig. 2a). The images in Fig. 2c display the evolution of the drop-sapphire interface from the bottom view, with the time identical to those in Fig. 2b. The light-gray regions indicate the existence of vapor between the drop and the substrate, while the dark gray regions represent the direct contact of the substrate with the drop[43]. In the entire process, the superhydrophilic pattern is constantly wetted by the drop, exhibiting uniform dark gray. By comparison, in stage I ($t = 1$ s), there bubbles exist around the superhydrophilic pattern (Supplementary Movie 3). These irregular bubbles are discontinuous and isolated by the solid–liquid contact regions. As time proceeds, the bubbles grow up and gradually coalesce[44]. At the following stage, a continuous vapor film forms and is divided into separated parts by the superhydrophilic pattern, indicating that the superhydrophobic regions turn Leidenfrost state. Meanwhile, the vapor film slightly elongates and starts to spin. At stage III, the continuous vapor film elongates, and two flows consistent with the spinning direction, which induce bulges at the two sides of the pattern, are observed to exhaust from the corners of the pattern (Supplementary Fig. 5).

Therefore, we can conclude that the drop spinning occurs in a "compartmentalized Leidenfrost state", from which the drop levitates on the superhydrophobic region and contacts the superhydrophilic pattern. The superhydrophilic pattern segments the Leidenfrost film into two distinct parts, thus reshaping the vapor flow beneath the drop. The "liquid wall" on the superhydrophilic pattern produces vapor and prevents crossover leakage. Conversely, Leidenfrost films formed in the superhydrophobic region exclude vapor and contribute two outside flows. The viscous force from the exhausting vapor may contribute to a force couple and consequently spins the drop[25,42].

To confirm the hypothesis, we build a model to understand this phenomenon. The variation of drop mass with time is shown in Fig. 2e. During the rotation process (29–112 s), the drop evaporation rate $\dot{m}$ is about 1.2 mg s$^{-1}$, which is much larger than that heated through the complete Leidenfrost layer at the same temperature difference between the drop and the substrate (20 °C), that is 0.1 mg s$^{-1}$ (Supplementary Information). Hence, we can hypothesize that the vapor is mainly generated at the superhydrophilic pattern, and the exhausting vapor between the drop and the substrate levitates the drop. According to the conservation of mass, we have:

$$\dot{m} \approx 2\pi R U h \rho_v \qquad (1)$$

where $U$ and $\rho_v$ are the radial velocity of the vapor flow and the density of the vapor. $R$ and $h$ are the drop radius and the thickness of the vapor layer, as shown in Fig. 2d.

As $h$ is much smaller than $R$, the lubrication approximation is utilized to link the horizontal pressure gradient $\Delta P$ and $U$:

$$\frac{\triangle P}{R} \approx 12\mu_v \frac{U}{h^2} \qquad (2)$$

where $\mu_v$ is the viscosity of the vapor. For a drop large than the capillary length $l_c$, $\Delta P \approx 2\rho g l_c$, where $\rho$ is the density of water, and $g$ is

the gravitational acceleration. At 100 °C, the capillary length of water $l_c$ is about 2.5 mm, resulting in a pressure of 50 Pa under the drop.

The outward flow of the vapor from the corner of the superhydrophilic pattern generates a viscous force to the liquid that serves as the driving force to spin the drop. We simplify the segmented vapor film to a rectangle, and the driving force $F_d$ can be obtained by:

$$F_d \approx \frac{(\mu_v \dot{m})^{1/3}(\rho g l_c)^{2/3}}{\pi \rho_v^{1/3} R} d^2 \qquad (3)$$

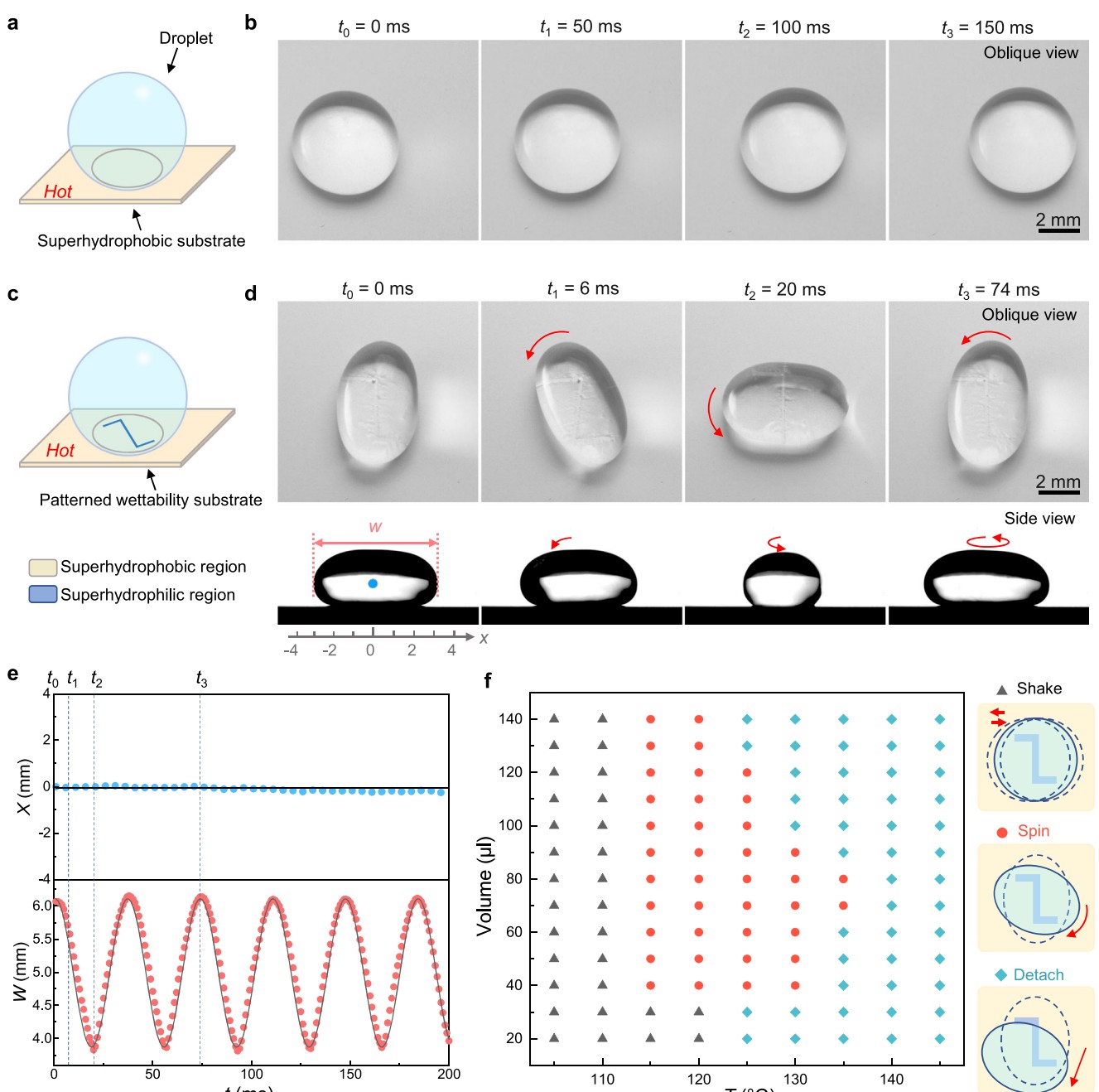

**Fig. 1 | Different vapor dynamics of drops on hot surfaces. a, b** Drop on a hot superhydrophobic surface. Benefitting from the beneath "intact" vapor film, a water drop almost "frictionless" wanders on the surface. **c** Schematic of a water drop sitting on a hot chemically heterogeneous surface. **d** Synchronized high-speed image sequences showing the drop spins on the chemically heterogeneous surface. The drop elongates and spins with a period of 74 ms. The red arrows indicate the rotation direction of the drop. **e** Width of the drop projection $W$ and

position of the drop centroid $X$ as functions of time $t$. $X$ remains almost unchanged, while $W$ varies periodically with time $t$. The period is 37 ms, which is half of the drop rotation period. The data are collected from the rotating process of (**d**). **f** Phase diagram summarizing the drop volume and the substrate temperature $T$ required for the drop spin. The temperature of the substrates in (**b**, **d**) is 120 °C, and the volume of the drops is 60 μl.

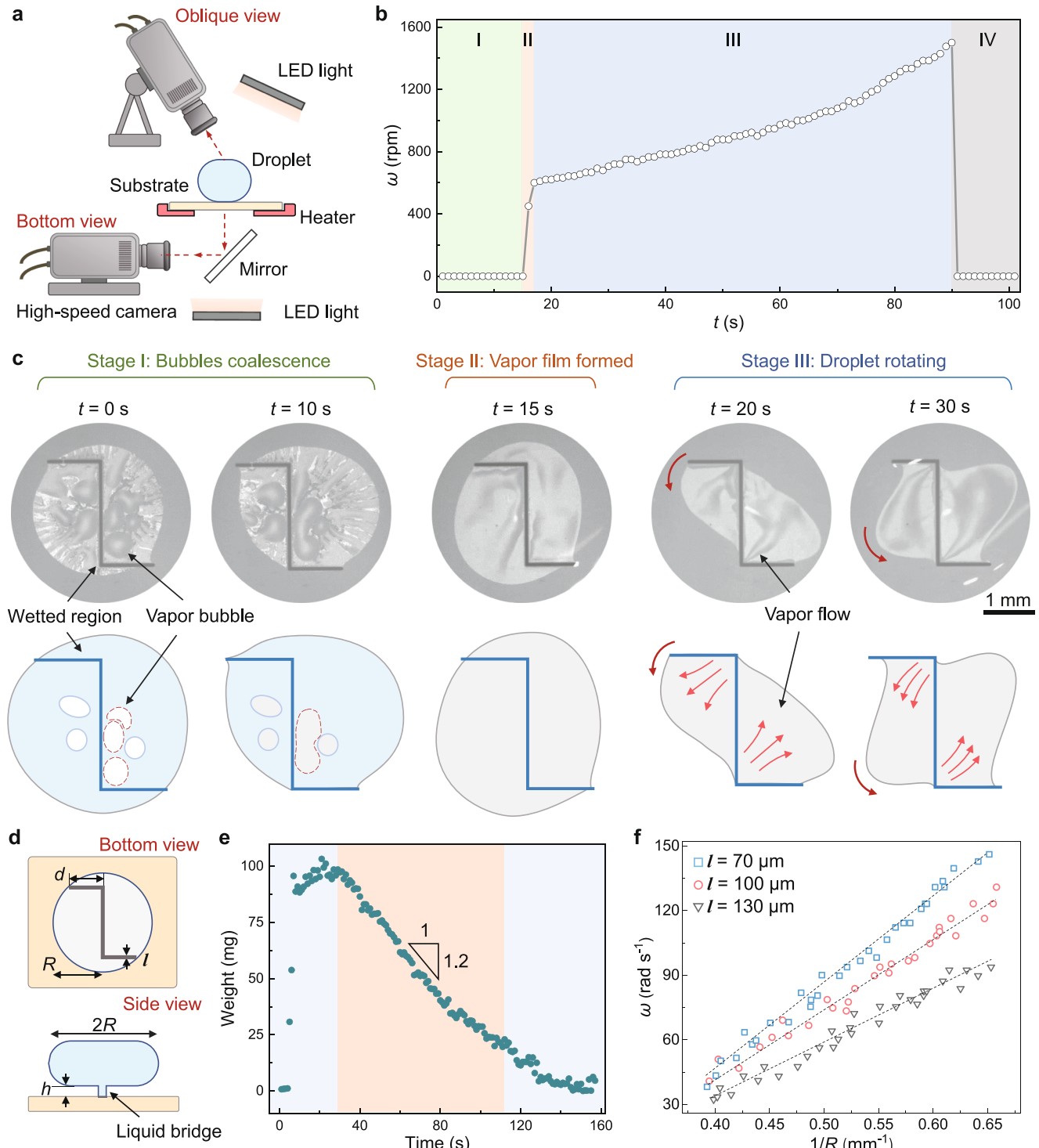

**Fig. 2 | Mechanism of drop spin by tailoring Leidenfrost film. a** Scheme of the drop observation facility. Synchronized high-speed cameras are used to record the drop evolution from oblique and bottom views. **b** Drop rotational speed $\omega$ as a function of time $t$. The whole process is divided into four stages. Stage I (0 s to 14 s): no obvious spin but a slight shake is observed. Stage II (15 s to 17 s): the drop starts to spin and the rotational speed rapidly increases. Stage III (18 s to 90 s): the drop spins rapidly, and the rotational speed gradually increases. Stage IV (91 s to 101 s): the drop spin ceases. When placed on the substrate, the volume of the drop is 80 μl. The temperature of the substrates is 120 °C. **c** Bottom views to show the drop evolution at different stages of I, II, and III in **b**. In the superhydrophobic regions, discontinuous bubbles exist at the bottom of the drop in stage I, while a continuous circular vapor film is formed in stage II. Then, the vapor film stretches into an oval in stage III. By comparison, in the entire process, the superhydrophilic pattern is well-wetted, meaning the Leidenfrost film is cut into segregative parts. The red arrows indicate the motion directions of vapor or drops. **d** Parameters describing the rotating drop from side and bottom views. **e** Mass change of the heated drop. The initial volume of the drop is 80 μl, and the temperature of the substrates is 120 °C. The rate of mass change during the rotating process (the orange region) is about 1.2 mg s$^{-1}$. **f** Drop rotational speed $\omega$ as a function of $1/R$. Dots of different colors are obtained experimentally, and the dashed lines are obtained based on theoretical analysis. Data for each width are collected from seven droplet rotation videos.

where $d$ is the side length of the superhydrophilic pattern, as defined in Fig. 2d. Since $F_d$ is considered to act on the corner of the super-hydrophilic pattern (Fig. 2c), we get the driving torque $M_d$:

$$M_d \approx 2 \frac{d}{\csc \pi/2} F_d \qquad (4)$$

Typical values for the parameters are: $\mu_v \approx 10^{-5}$ Pa s, $\dot{m} \approx 1.2 \times 10^{-6}$ kg s$^{-1}$, $\rho \approx 10^3$ kg m$^{-3}$, $g \approx 9.8$ m s$^{-2}$, $l_c \approx 2.5 \times 10^{-3}$ m, $\rho_v \approx 1$ kg m$^{-3}$, and $d \approx 1.5 \times 10^{-3}$ m. Hence, we find $F_d \approx 7 \times 10^{-7}$ N and $M_d \approx 1.5 \times 10^{-9}$ Nm with $R \approx 2 \times 10^{-3}$ m, which are consistent with the previous studies[26].

Meanwhile, velocity gradients exist in the drop and the liquid bridge that connects the drop and the substrate (Fig. 2d), causing viscous shear forces to resist the drop rotation. As the velocity gradient in the drop thickness direction is about 1/4 of the rotational speed (Supplementary Fig. 6), the resisting torque within the drop is: $M_r' = \pi \mu \omega R^4 / 16 l_c$, which yields a value of $10^{-10}$ Nm that is much smaller than the driving torque. By comparison, the resisting torque within the liquid bridge is:

$$M_r \approx \frac{8}{3} \frac{\mu \omega l d^3 (\rho g l_c)^{1/3}}{(\mu_v \dot{m})^{1/3}} \qquad (5)$$

demonstrates a value of about $1.1 \times 10^{-9}$ Nm, which is comparable with the driving force. Therefore, considering the driving and resisting torques are balanced in a spinning drop, the drop rotational speed can be calculated by:

$$\omega \sim a l^{-1} R^{-1} \qquad (6)$$

where $a = (\mu_v \dot{m})^{2/3} (\rho g l_c)^{1/3} / 10 \mu \rho_v^{1/3}$. For a drop with a radius of 2 mm, $\omega$ is at a scale of 100 rad s$^{-1}$, which coincides with experimental data.

Further, we measure the drop rotational speed $\omega$ and plot its dependence on the equatorial radius $R$ in Fig. 2e to verify the model. Superhydrophilic patterns with different widths are adopted in the tests, as demonstrated by the different colored dots. The rotational speed $\omega$ shows a linear relationship with $R^{-1}$, while at a fixed $R^{-1}$, $\omega$ is negatively correlated to the width of the superhydrophilic pattern $l$, which is in accordance with our model. Details of the model are provided in Supplementary Information.

According to the above model, we can interpret the drop behavior shown in the phase diagram of Fig. 1f. When the temperature of sub-strate is below 115 °C, the continuous vapor film cannot form in the superhydrophobic region and a large adhesion exists. However, the driving force from the outside flows is relatively small due to the weak vaporization. Consequently, the drop cannot spin but display a slight shake. With the increase of the temperature, the adhesion decreases because of the formation of the vapor film, and the driving force increases contribute to the enhanced vaporization and the accelerated vapor flow. Once the driving force is larger than the resisting force, the drop begins to spin. However, if the temperature is above 135 °C, extremely violent vaporization happens at the superhydrophilic pat-tern, which generates a large driving force, making the super-hydrophilic pattern cannot seize the spinning drop.

## General principle of drop vapor regulation

The drop gyro originates from the symmetry-breaking vapor flows that is realized by tailoring the Leidenfrost film. Therefore, we can modify the configuration of the Leidenfrost film by designing super-hydrophilic patterns to rectify the vapor dynamics, and realize diverse drop manipulation[45]. Figure 3 summarizes the correlation between the symmetry of the superhydrophilic pattern and the drop manner (Supplementary Movie 6). Generally, a superhydrophilic pattern with central symmetric can split the Leidenfrost film into several segments with the same area and symmetry, which may result in the force couple to rotate a drop, as exampled in Fig. 3a. By contrast, the separated vapor films by an axial symmetry pattern, displays the same symmetry but different areas, generating a directional net force on the drop. Therefore, the drop on the substrate with an angle-shaped pattern is unilateral stretched to the opening direction and exhibits an egg shape (Fig. 3b). Combining the central and axial symmetry, the split vapor film contributes to well-balanced viscous drags, making the drop symmetrical elongate and remain stationary, as displayed in Fig. 3c. Finally, the pattern with asymmetric patterns will produce a torque consisting of unbalanced and unequaled forces, which induces the unstable spinning drop (Fig. 3d). More instances of the super-hydrophilic patterns for drop actuation are provided in Supplementary Fig. 7.

## Applications of drop vapor regulation

The unique drop vapor manipulation strategy can be harnessed for many applications. The steam-driven engine plays an indispensable role in mechanics driving, ship navigation, and power generation, while its utilization in micromachines is rather limited. One reason is that the vapor-guided components, like turbines, cranks, and pistons, indubitably complicate the apparatus and make manufacturing difficult[46]. Extensive works have been carried out to manufacture heat engines using the "complete" Leidenfrost effect[47–51], in which the vapor is controlled by the physical structure. Using the flat chemically het-erogeneous surface, vapor can be regulated by the "compartmenta-lized Leidenfrost effect", which provides an optional strategy for vapor-to-mechanical conversion. We design and display a "droplet steam engine" with a size of 1 cm, and a weight of 217 mg, as displayed in Fig. 4a. The droplet steam engine has a simple architecture, con-taining a heater with a chemically heterogeneous surface, a gear, and a power transmission part to connect the drop and the gear (Fig. 4b). When a 100 μl drop is injected into the engine, the drop is heated and starts to spin (Fig. 4c). The curve of rotating speed versus time, dis-played in Fig. 4d, shows that a single drop can actuate the gear to rotate continuously for nearly 50 s with the maximum speed exceed-ing 200 rpm (Supplementary Movie 7). The droplet steam engine can be further simplified by directly placing an airscrew on the drop. The airscrew can consistently and steadily rotate for more than 50 s with-out any supporting apparatus (Supplementary Fig. 8 and Supplemen-tary Movie 8). Furthermore, the wettability patterns can be prepared by a spraying method, which is easier to be prepared on thin and flexible films or fragile surfaces (Supplementary Fig. 9).

This Leidenfrost vapor manipulation strategy employs chemical heterogeneity, which shows promising applications in special and harsh circumstances, such as limited space. Figure 4e–h displays the driving of a shaft by filling water in a narrow gap. As shown in Fig. 4e, a water outlet connecting with an injection pump is designed at the center of the top shaft, with a heated bottom pillar decorated with a wettability pattern on its top. When water is injected into the gap, the wettability pattern shapes the vapor, actuating the drop rotation to drive the shaft (Fig. 4f and Supplementary Movie 9). Figure 4g demonstrates the drop actuator in a narrow gap of 1 mm. The actua-tion of the top shaft can be efficiently controlled by water injection in the gap (Fig. 4h). Besides mechanical driving, the drop gyro can also be used to waste heat-based electricity generation and nanoparticle synthesis, as explored in Supplementary Figs. 10 and 11. In addition, directional droplet transport can be achieved by altering the wett-ability pattern from central symmetric to axially symmetric. As shown in Fig. 4i, j, when placed on the substrate with a "herringbone" wett-ability pattern[52,53], a drop moves along the pattern with a speed of 4 cm s$^{-1}$ (Supplementary Movie 10), which is comparable with the Lei-denfrost drops on micro-patterned electrodes[54]. In addition, the drop can even be transported to anti-gravity, as demonstrated by tilting the substrate to 5° (Fig. 4k and Supplementary Movie 11).

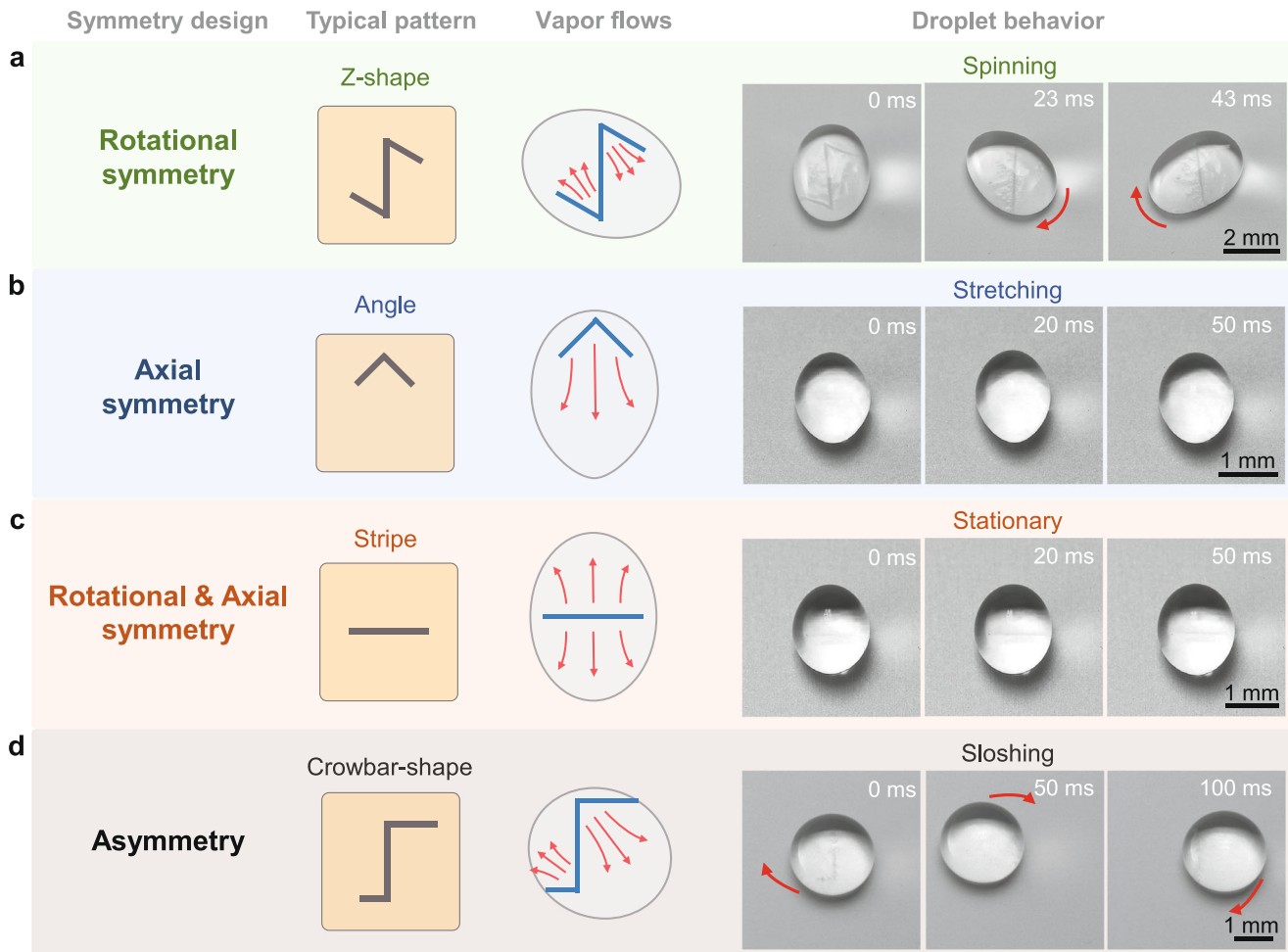

**Fig. 3 | General principle of chemical heterogeneity design for diverse drop vapor regulation. a** Drop with an ellipsoid shape spins on the hot super-hydrophobic surface with a Z-shaped pattern that displays central symmetric. **b** When placed on the angle patterned surface that has axial symmetry, the uni-directional vapor flow causes the drop to unilateral stretch, and exhibits an egg shape. **c** "Stript" pattern that combines the central and axial symmetry contributes to balanced vapor flows, making a stationary drop with a symmetrical olive shape. **d** A spherical shape drop sloshes on a crowbar-shaped pattern surface that is asymmetric, due to the differentiated vapor flows. The temperature of the surfaces is 120 °C. The initial volumes of drops in (**a**, **d**) are 60 μl, and in (**b**, **c**) are 10 μl. The red arrows indicate the motion directions of vapor or drops.

In summary, we report a unique strategy to tailor the Lei-denfrost film for precise droplet vapor rectification by designing chemical heterogeneity on a planar surface, and reveal a general principle for controlling droplet vapor by contriving the symmetry of the wettability patterns. Moreover, we demonstrate its broad applications in mechanical drives and microfluidics. Bene-fiting from the "compartmentalized Leidenfrost state" achieved by the wettability pattern, this strategy is effective in a specific "low" operating temperature interval, which implies that it may have a higher energy conversion efficiency compared to the "normal Leidenfrost effect". This finding opens a new avenue to control the Leidenfrost dynamics, and enriches the knowledge between fluid dynamics and surface science, which shows great potential in micromechanics and vapor energy utilization.

## Methods
### Materials
Porous aluminum plates were purchased from Beijing NanoThink Printing Co. Sapphire plates were purchased from Donghai County Alfa Quartz Products Co. Sodium citrate, auric acid, and 1H,1H,2H,2H-perfluorodecyltrimethoxysilane (PFOTS) were purchased from J&k Scientific Co. Ethanol was purchased from Tianjin Concord Technol-ogy Co. Ultrapure water with electrical resistivity greater than 18 MΩ cm was used in all experiments, supplied by Milli-Q Advantage A10 (Millipore). Glaco Mirror Coat Zero was purchased from Soft99 Co.

### Preparation of the heterogeneous surface
Two kinds of materials, porous aluminum and sapphire, were used to prepare heterogeneous surfaces, as shown in Supplementary Fig. 1. Porous aluminum plates were cleaned with ethanol, acetone, deio-nized water, and blow-dried with nitrogen sequentially. Surface-modification with PFOTS by chemical vapor deposition (CVD) at 90 °C for 6 h made the porous aluminum plates superhydrophobic (Supplementary Fig. 1). Then the surfaces were covered with photo-masks and exposed under a UV light (400 W, 365 nm) for 8 h. After exposure, the uncovered regions became superhydrophilic. Unless indicated, the experiments were performed on the porous aluminum plates. For the bottom visualization experiment, sapphire plates were employed due to the high thermal conductivity and light transmit-tance. To make sapphire plates superhydrophobic, we slowly dipped the cleaned plates into the Glaco solution and then pulled them out. After drying at room temperature for 10 minutes, the contact angle of the sapphire plates reached 154 ± 1° (Supplementary Fig. 1). The superhydrophilic patterns on the superhydrophobic sapphire plates were prepared using a process similar to that treating the aluminum plates.

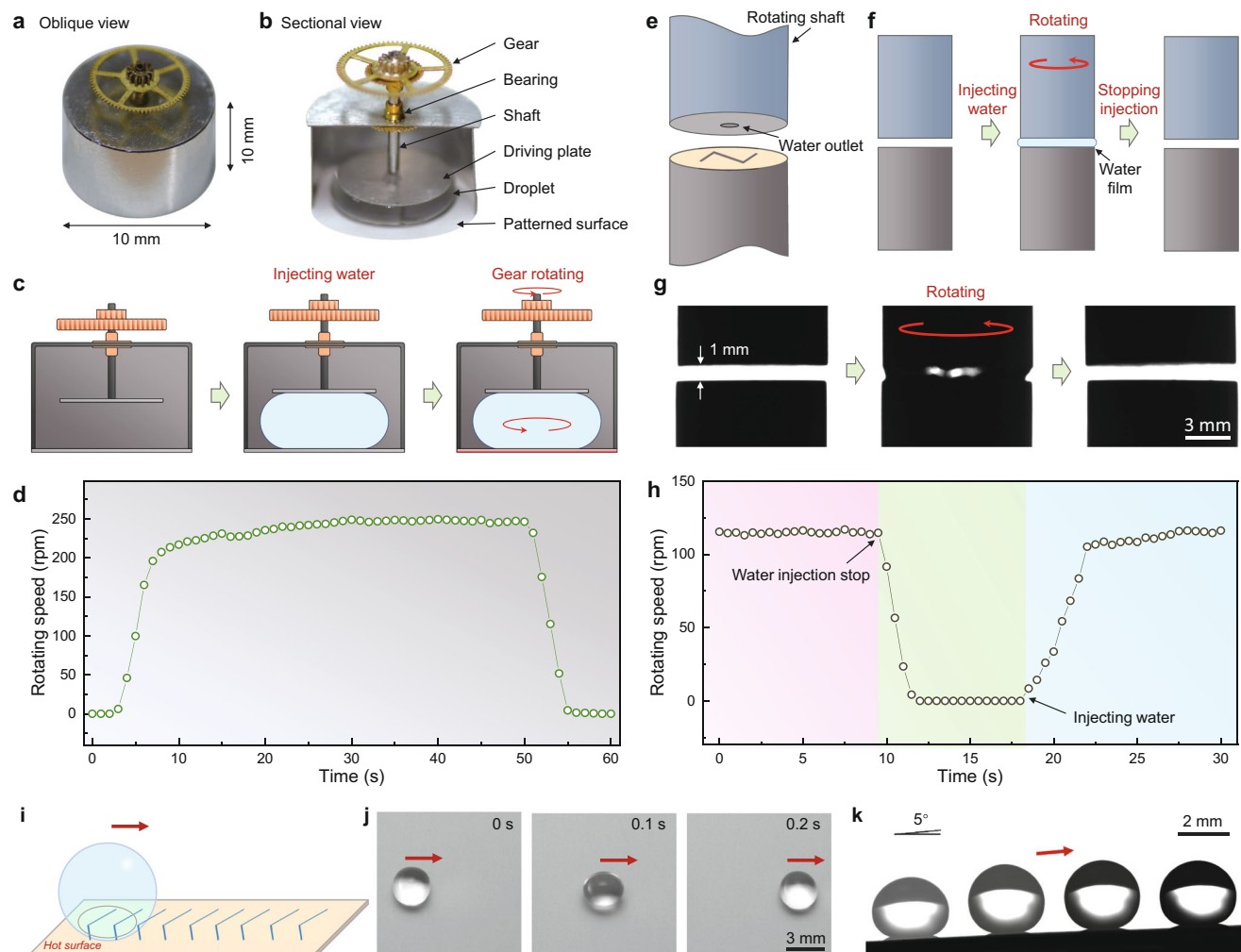

**Fig. 4 | Applications of the tailoring Leidenfrost film. a–d** Droplet steam engine. **a**, **b** Photographs of the droplet steam engine. The droplet steam engine consists of a heater with a chemically heterogeneous surface, a gear, and a force transmission device to connect the drop and the gear. **c** Working process of the droplet steam engine. When water is added, the generated vapor actuates the water drop rotating, which in turn drives the steam engine to work. **d** Curve of the steam engine speed as a function of time. After adding a 100-μl drop, the steam engine can rotate for nearly 40 s. **e**–**h** Shaft driving in a narrow gap. **e** Structure of the shaft driving device. **f** Illustration of the shaft driving strategy. Water injection results in a torque acting on the upper shaft, and thus rotates the shaft. **g** Images of the shaft driving process. The distance between the shaft and the heating pillar is 1 mm. The shaft starts to rotate after the injection of water (the middle image), while the rotation ceases when stopping the injection (the right image). **h** Rotating speed of shaft versus time. The rotating speed of the shaft is about 130 rpm, which decreases to zero within 5 s after stopping water injection (9 s), while re-injecting water (16 s) causes the shaft to return to about 130 rpm within 8 s. The red arrows indicate the rotation directions of the shaft. **i**–**j** Directional transport of drops using a planar hot surface. **i** Illustration of the directional transport of a drop on a superhydrophobic surface with a "herringbone" superhydrophilic pattern. **j** Time-lapse trajectory of a drop on the hot pattern surface. **k** Anti-gravity propulsion of a drop on a 5-degree inclined surface. The temperature of the surfaces is 135 °C. The volume of drops in (**j**, **k**) is 60 μl. The red arrows indicate the motion directions of the drops.

## Recording of the drop spin

The drop-spinning processes were recorded by high-speed cameras. The recording speed was set as 3000 frames per second (fps). The oblique view, side view, and bottom view were recorded by Phantom V12.1, Phantom VEO401L, and Photron Fastcam NOVA S12, respectively. The oblique view and side view were recorded by placing drops on the porous aluminum plates, while the bottom view was obtained by placing drops on the sapphire plates.

## Wettability characterization

The contact angle, the advancing contact angle, and the receding contact angle were measured by OCA 25 (Dataphysics). A sessile drop with a volume of 2 μl was placed on the surface to measure the static contact angle. The advancing contact angle and the receding contact angle were obtained by injecting water into and sucking water from a 2-μl droplet that was placed on the surface. The flow rates of the injecting and sucking processes were

$0.2\,\mu l\,s^{-1}$. All the data was averaged from independent measurements of five samples.

## Microstructure characterization

Scanning electron microscopy (SEM) images of the porous aluminum plates and the Sapphire plates were obtained on JEOL-F7500 at 10 kV.

## Preparation of the droplet steam engine

The droplet steam engine consisted of four major components: an actuation part consisting of a drop and a patterned wettability surface, a transmission part consisting of a driving plate and a shaft, a framework, and a gear, as shown in Fig. 3a and Supplementary Fig. 3. The diameter and height of the micro engine were 10 mm and 12 mm, respectively. The drop supported the transmission part and the gear by surface tension. When spinning, the drop transferred the torque to the driving plate. To float the driving plate and the transmission part, special surface modifications were required for the driving plate in

addition to minimizing the weight of these parts. The upper surface of the driving plate was superhydrophobic with low adhesion (sprayed by the Glaco solution), while the lower surface was hydrophobic with high adhesion. Therefore, we could evaluate the maximum load of the drop, that is $\pi \gamma D/g \approx 138$ mg (where $\gamma$ is the surface tension of water, $D$ is the diameter of the driving plate, and $g$ is the gravitational acceleration). In our study, the driving plate of 13.7 mg was obtained by cutting an aluminum sheet of 50 μm in thickness using a laser cutting machine. The shaft was made of steel, and the diameter, height, and weight were 1 mm, 8 mm, and 32.4 mg, respectively. A bearing was mounted on the framework to reduce friction and lower the vibration of the shaft during the rotation process. The gear with a diameter of 8 mm was made of copper, and the weight is 51.8 mg.

### Shaft driven by the spinning drop
A shaft is suspended above a heating pillar with a diameter of 10 mm. A 1 mm diameter outlet that is connected to a syringe pump is designed in the center of the shaft. The distance between the lower surface of the shaft and the upper surface of the heating pillar is 1 mm. A syringe pump (Shenchen SPM-ZU-I) is used to inject water into the narrow gap at a rate of 115 μl min$^{-1}$. The temperature of the heated column is 125 °C.

## Data availability
The authors declare that the data supporting the findings of this study are available within the paper and its supplementary information files. Source data are provided with this paper. Source data are provided with this paper.

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

## Acknowledgements

We thank Jianing Wu from Sun yat-sen university for the discussions. This
work is supported by the National Key R&D Program of China
(2018YFA0208501 (Y.S.)), the National Natural Science Foundation of
China (22272182 (H.L.), 52293473 (H.L.), 51903240 (H.L.), 91963212
(Y.S.), 11988102 (C.S.)), Beijing National Laboratory for Molecular Sci-
ences (BNLMS-CXXM-202005 (Y.S.)), and the Youth Innovation Promo-
tion Association of CAS (2023039 (H.L.)).

## Author contributions

A.L., H.L., C.S., and Y.S. conceived the project. A.L. and H.L. per-
formed the experiments. A.L., H.L., S.L., C.S., and Y.S. analyzed the
data and discussed the results. A.L., H.L., S.L., C.S., and Y.S. wrote
and proofread the paper. All authors made comments and
approved the manuscript.

## Competing interests

The authors declare no competing interests.

## Additional information

**Supplementary information** The online version contains
supplementary material available at

Huizeng Li, Chao Sun or Yanlin Song.

**Peer review information** *Nature Communications* thanks the anon-
ymous reviewers for their contribution to the peer review of this work. A
peer review file is available.

