## [Peer Review File · Nature Communications]

REVIEWER COMMENTS

Reviewer #1 (Remarks to the Author):

Report on the paper by Li et al.

In this paper, Li et al report that a Leidenfrost drop (not droplet, by the way, all drops are quite large) can be actuated (shaking, rotating, detaching) by chemical defects on the substrate. This looks paradoxical since Leidenfrost implies levitation, and levitation implies no-sensitivity to the substrate, but the authors work in a narrow interval of temperature where the drop can both levitate on a hot, hydrophobic solid and contact its hydrophilic defects.

Well, this is an original, esthetic study. It opens a field that we could call “compartmentalized Leidenfrost effect” and warrants, in my opinion, a publication in an excellent journal such as Nature Communications. However, I also feel that the paper suffers from a few missing things, and I enclose below a list of remarks or criticisms. I would be happy to read a revised version of this study before I can make a final recommendation.

Main points:

1. The observed effects are related to the possibility of having a mixed state, that is, wetting on the defect, and levitating on the rest of the substrate. Hence, we expect that it should be only found in a relatively narrow domain of temperature, which is indeed the case in fig 1f. I think the authors should be clearer about this fact, and discuss more extensively that they need both levitation on the hydrophobic solid (temperature large enough) and contact on the defect (temperature not too large), which fixes the interval of temperature (and its narrow character) in which the effects take place.

2. It is interesting that something happens below the boiling point of water (shake mode in fig 1f). This should be commented, and a qualitative explanation should be given. The period of shaking should be given, and maybe compared to the “natural” time of oscillation of a drop. Note that I am not asking here for a quantitative model.

3. In the model (lines 12 to 16 on page 10), justify why you use force balance and not torque balance (the object is rotating). More importantly, assuming for instance that the torque is applied on the same distance d so that force and torque balances provide the same scaling, the force balance itself is not obvious. Firstly, if I follow the authors, they eliminate U by using the formula on line 11, but ΔP being constant (its value, by the way, is not given), and h being proportional to square root of R , U is independent of R so that ω is independent of R . Secondly, I am not fully confident that I should follow the authors. At least, they should carefully justify the scaling laws on lines 13 and 14. For instance, it is not obvious that U on line 13 is the U of line 11: in their system, the production of vapor mainly occurs at the hydrophilic line, so that U is fixed by boiling at this line, and not by line 11, established without contact and boiling. U in line 13 would just be a mean velocity of vapor produced by the line, and not explicitly given by an equation. Similarly, it is not obvious that the distance to be taken into account in the gradient of velocity on line 14 is h . In any case, the authors should give more details (much more) about the model, even if they remain at a scaling level – I should add, particularly if they remain at a scaling level.

If the authors can justify their final formula, it seems that the comparison of the model works for the scaling in R . However, the authors should calculate the coefficient from the scaling law and check that it provides the correct order of magnitude for the coefficient deduced from the data.

4. Fig 4i,j,k: The fishbone pattern is extremely reminiscent of that proposed by Soto et al. (PRF, 2016), which, curiously, is not cited. I think it should, because the principles described in Soto's paper are close to the ones exploited by the authors. In addition, the authors should compare the dynamics they obtain to the one reported by Soto.

At least as surprising, the work by the authors published in Nature Communications (2019) is not cited either although it deals with very similar ideas, that is, conversion of energy into rotation using chiral patterns. Here again, it must be cited, for instance in the introduction, or when describing the rotation of the drop.

Minor details:

1. Sentences that exaggerate the applications of an effect always weaken a paper, and we have a few examples of this kind in the paper. For instance, I would suppress the sentence "and demonstrate the potential of the droplet steam engine..." in the abstract. These drops are very, very poor engines when you compare the injected energy and the resulting kinetic energy...

2. Rank the ref. 5 to 7 in the chronological order. It is important to credit who had the ideas first. Same thing for ref. 11 to 14.

3. Line 21 on page 5: is it 4π in the formula or 2π ?

4. Line 9 on page 10: a verb is missing. Reword also the next sentence.

5. Line 12 on page 11: dargs should be corrected.

Reviewer #2 (Remarks to the Author):

Language is impenetrable, with inappropriate synonyms used instead of more appropriate words. The abstract needs a thorough re-work and language check. Examples of unusual wording below, however the language needs revising throughout:

- Page 1 Line 15-16: "The vapour escaping from the Leidenfrost film causes riotous flows, and moves the droplet to wander around"
- Line 21: "the phase-change vapour dynamics remains fragmentary"
- Page 3 Line 7: "tranquilized boiling"
- Line 12: "explosive boiling"
- Throughout: "wandering"
- Page 5 Line 6: "ramble"
- Page 12: Lines 1-2: "The droplet gyro originates from the force couple induced by symmetry-breaking vapour flows that is maneuvered through tailoring the Leidenfrost film".

Equations need to be separated on placed on dedicated lines, rather than inline for a number of defined parameters, for example:

- Page 10 Line 5: separate the equation as parameter definition is unclear

Extensive works by McHale et al with multiple institutions (Nottingham Trent, Northumbria, Durham and Edinburgh) in the areas of the Leidenfrost propulsion and control (selective and bulk), sublimation heat engine, hydrophobicity, coatings and varying surface wettability (G-SLIPS) have not been referred to at all, which is a glaring omission which must be rectified. The literature would not be complete without this information being included.

The paper has some merit but requires a full review and comparison with existing literature and the language used makes it difficult to read.

Reviewer #1 (Remarks to the Author):

Report on the paper by Li et al.

In this paper, Li et al report that a Leidenfrost drop (not droplet, by the way, all drops are quite large) can be actuated (shaking, rotating, detaching) by chemical defects on the substrate. This looks paradoxical since Leidenfrost implies levitation, and levitation implies no-sensitivity to the substrate, but the authors work in a narrow interval of temperature where the drop can both levitate on a hot, hydrophobic solid and contact its hydrophilic defects.

Well, this is an original, esthetic study. It opens a field that we could call “compartmentalized Leidenfrost effect” and warrants, in my opinion, a publication in an excellent journal such as Nature Communications. However, I also feel that the paper suffers from a few missing things, and I enclose below a list of remarks or criticisms. I would be happy to read a revised version of this study before I can make a final recommendation.

Response: We are deeply thankful to the reviewer for the time and effort in reviewing our manuscript and providing a concise and accurate summary. Moreover, we are very grateful for the high evaluation of our work. According to the reviewer's insightful comments, we have revised the manuscript and made a point-to-point response.

- (1) We have thoroughly checked and replaced the word “droplet” to “drop” in the manuscript and the supplementary materials.
- (2) We have performed necessary experiments and quantified the drop shaking periods at low temperatures, as well as explained the mechanism.
- (3) Following the reviewer’s suggestion, the model of drop rotation has been modified.
- (4) The overlooked references and the relevant discussion have been added accordingly.
- (5) We have suppressed the part of the application and modified the Abstract.
- (6) The manuscript has been thoroughly examined, and the typos and the order of references have been revised.

Main points:

1. The observed effects are related to the possibility of having a mixed state, that is, wetting on the defect, and levitating on the rest of the substrate. Hence, we expect that it should be only found in a relatively narrow domain of temperature, which is indeed the case in fig 1f. I think the authors should be clearer about this fact, and discuss more extensively that they need both levitation on the hydrophobic solid (temperature large enough) and contact on the defect (temperature not too large), which fixes the interval of temperature (and its narrow character) in which the effects take place.

Response: Thanks for the constructive comments. We agree with the reviewer that the drop rotation occurs in a “compartmentalized Leidenfrost state”, in which contact boiling occurs at superhydrophilic pattern and Leidenfrost boiling happens at superhydrophobic region.

To make the fact clearer, the sentence “The superhydrophilic region directly contacts the drop and vaporizes the water, while a vapour film is formed on the superhydrophobic surrounding to exhaust vapour and suppress heat transfer.” is added in the Abstract, as well as the sentence “Therefore, we can conclude that the drop spinning occurs in a ‘compartmentalized Leidenfrost state’, in which the drop levitates on the superhydrophobic region and contact on the superhydrophilic pattern.” on Page 6.

We have added a paragraph “According to the above model, we can interpret...” on Page 8 to discuss the effect of temperature on the drop behaviour.

2. It is interesting that something happens below the boiling point of water (shake mode in fig 1f). This should be commented, and a qualitative explanation should be given. The period of shaking should be given, and maybe compared to the “natural” time of oscillation of a drop. Note that I am not asking here for a quantitative model.

Response: Thanks for the constructive comments. We have performed additional experiments to study the drop shaking. In these experiments, drops with the same volume (50 μL) were placed on substrates at different temperatures, and the tests at each temperature were repeated five times. We found that the shaking direction is irregular even at the same temperature, and the drop shaking is difficult to detect when temperature below 100 $^{\circ}\text{C}$. The shaking period between 100 $^{\circ}\text{C}$ to 110 $^{\circ}\text{C}$ was calculated and shown in Fig. R1, where we could find that the period was larger at a lower temperature and smaller at a higher temperature, and was different from the “natural” time of drop, that is $t \approx 2.2\tau \equiv 2.2 \cdot \sqrt{\frac{\rho R^3}{\gamma}} \approx 60 \text{ ms}$ (Rayleigh L. *Proc. R. Soc. Lond.* 1987, 29, 71; Richard D., Clanet C. and Quéré D. *Nature* 2002, 417, 811).

Figure R1. Period of the drop shaking.

The possible reason for the shaking is that at temperatures below the Leidenfrost boiling point, the vapour film in the superhydrophobic region is not completely formed and a large adhesion resistance exists, while the vaporization in the superhydrophilic region is weak and the driving force from the vapor flow is small. As a result, the drop does not rotate but slightly shakes on the substrate. The velocity of airflow becomes larger when the temperature rises, contributing to an increased shaking frequency.

We have added the related discussion of “When the temperature of substrate is below 115 $^{\circ}\text{C}$, the continuous vapour film cannot form in the superhydrophobic region and a large adhesion exists. However, the driving force from the outside flows is relatively small since the weak vaporization. Consequently, the drop cannot spin but display a slight shake.” on Page 8.

3. In the model (lines 12 to 16 on page 10), justify why you use force balance and not torque balance (the object is rotating). More importantly, assuming for instance that the torque is applied on the same distance d so that force and torque balances provide the same scaling, the force balance itself is not obvious. Firstly, if I follow the authors, they eliminate U by using the formula on line 11, but ΔP being constant (its value, by the way, is not given), and h being proportional to square root of R , U is independent of R so that ω is independent of R . Secondly, I am not fully confident

that I should follow the authors. At least, they should carefully justify the scaling laws on lines 13 and 14. For instance, it is not obvious that U on line 13 is the U of line 11: in their system, the production of vapor mainly occurs at the hydrophilic line, so that U is fixed by boiling at this line, and not by line 11, established without contact and boiling. U in line 13 would just be a mean velocity of vapor produced by the line, and not explicitly given by an equation. Similarly, it is not obvious that the distance to be taken into account in the gradient of velocity on line 14 is h . In any case, the authors should give more details (much more) about the model, even if they remain at a scaling level – I should add, particularly if they remain at a scaling level.

If the authors can justify their final formula, it seems that the comparison of the model works for the scaling in R . However, the authors should calculate the coefficient from the scaling law and check that it provides the correct order of magnitude for the coefficient deduced from the data.

Response: We appreciate the reviewers for the professional comments, and we agree with the reviewer that the torque balance is more appropriate to analyze the rotation than the force balance. Therefore, we have revised the model on Page 6 and 7.

In our mode, we assume that vapour is generated mainly from the superhydrophilic pattern, and the exhausting vapour between the drop and the substrate levitates the drop, since the heat conductive coefficient of water multiplied by the area of the superhydrophilic pattern (at the scale of 10^{-6}) is much larger than that of vapour multiplied by the area of the drop bottom (at the scale of 10^{-7}).

For a high-speed rotating drop, heat is considered to be transferred by convection, and the heat transfer distance between the substrate and the drop is the thickness of thermal boundary layer.

As the Prandtl number of water is about 1 near 100 °C, the thickness of the thermal boundary layer is comparable to the thickness of the hydrodynamic boundary layer δ . Hence, $\delta_t \approx \delta \sim \sqrt{t\mu/\rho}$, in which $t \sim l/\omega R$ is the development time of the flow boundary layer (ω is the rotational speed of the drop), μ and ρ are the viscosity and density of water. Therefore, we have the water vapourization rate \dot{m} :

$$\dot{m} \sim \frac{k\Delta T l d}{L\delta_t} \quad (1)$$

where k is the thermal conductivity of water, ΔT is the temperature difference between the substrate and the water, L is the latent heat of water vaporization, and δ_t is the thickness of the thermal boundary layer.

Typically, the radial velocity of vapour flow U is lower than 1 m s^{-1} , much smaller than the sound velocity. Thus, the vapour in the vapour layer can be treated as incompressible. According to the conservation of mass, the water vapourization rate scales as the vapour exhaust rate:

$$\dot{m} \sim \rho_v h U R \quad (2)$$

where ρ_v is the density of the vapour, and h is the thickness of the vapour layer. Since $h \ll R$, the lubrication approximation is utilized to link the horizontal pressure gradient and U :

$$\frac{\Delta P}{R} \sim \mu_v \frac{U}{h^2} \quad (3)$$

where ΔP is the vapour pressure imposed by the drop, and μ_v is the viscosity of the vapour. For the drop large than the Capillary length l_c , $\Delta P \sim \rho g H$, where H is the drop height, and $H \sim l_c \sin \theta$. Therefore, we get

$$\Delta P \sim \sqrt{\rho g \gamma} \quad (4)$$

when considering a contact angle θ of 160°.

Combining equations (1), (2) (3), and (4), we get h and U :

$$h \sim \left(\frac{k\Delta T d \mu_v}{\rho_v L} \right)^{1/3} \left(\frac{\omega l R}{\mu g \gamma} \right)^{1/6} \quad (4)$$

$$U \sim \left(\frac{k\Delta T d}{\rho_v L} \right)^{2/3} (\rho)^{1/2} (g\gamma)^{1/6} \left(\frac{\omega l}{\mu \mu_v} \right)^{1/3} R^{-2/3} \quad (5)$$

Here, the viscous force can be obtained by $F_d \sim \mu_v \frac{U}{h} A$, where A is the area of the vapour film.

We can further simplify this area to be a sector, thus $A \sim d^2$. Therefore, the viscous force is:

$$F_d \sim \left(\frac{\mu_v k \Delta T d g \gamma}{\rho_v L} \right) (\rho)^{1/2} (d)^{7/3} \left(\frac{\omega l}{\mu} \right)^{1/6} R^{-5/6} \quad (6)$$

Typical values for the parameters are: $\mu_v \approx 2 \times 10^{-5}$ Pa s, $\rho \approx 10^3$ kg m⁻³, $\rho_v \approx 1$ kg m⁻³, $g \approx 9.8$ m s⁻², $\gamma \approx 7.2 \times 10^{-2}$ N m⁻¹, $\Delta T \approx 30^\circ\text{C}$, $L \approx 10^6$ J kg⁻¹, $k \approx 0.7$ W m⁻¹K⁻¹, $l \approx 10^{-4}$ m, $d \approx 1.5 \times 10^{-3}$ m, $\omega \approx 800$ rpm, and $R \approx 3 \times 10^{-3}$ m. Hence, we find that the driving force F_d is at the scale of 10^{-5} N.

As shown in Fig. 2b and Supplementary Movie 3, the vapour flows out from the corner of the superhydrophilic pattern. Hence, we assume that the driving force acts on the corner, which contributes to the driving torque M_d is:

$$M_d = 2 \sin \frac{\pi}{2} d \cdot F_d \quad (7)$$

Then, we will calculate the resisting torque M_r . Three major forces, including the adhesion force from the substrate, the air drag, and the viscous shear force, are usually considered to resist the drop rotation. For a levitating drop, the adhesion force can be neglected.

For the drop with a radius of 3 mm and a rotational speed of 800 rpm, the linear velocity at the edge is $U_1 \approx 0.3$ m s⁻¹. The Reynolds number $Re = 2\rho_v R U_1 / \mu_v$ is at the scale of 10^2 . Therefore, a boundary layer develops around the drop with a characteristic thickness δ' , and the resisting force of the drop movement is the viscous drag in the boundary layer. Since $\delta' \sim \sqrt{\nu_v t}$, we can get this drag force:

$$F_{\text{air drag}} \sim \sqrt{\rho_v \mu_v} (R U_1)^{3/2} \quad (8)$$

here $\nu_v = \mu_v / \rho_v$ is the kinematic viscosity and t is the time of development of the boundary layer, naturally scales as R/U_1 . With $R \approx 3$ mm, we find $F_{\text{air drag}}$ is at the scale of 10^{-7} N, much smaller than the driving force.

For the levitating drop, a liquid bridge connecting the drop and the substrate is formed on the superhydrophilic region. Velocity gradients exist in the liquid bridge, resulting in a viscous shear force that resists the drop spin. As shown in Fig. R2, the superhydrophilic pattern can be split into two parts (I and II).

Fig. R2. Scheme of the resisting torque.

If we simplify pattern II as an arc, the resisting torque M_r is:

$$M_r = \oint dF \cdot r \sim \int_0^d \mu \frac{\omega r}{h} l \cdot r dr + \mu \frac{\omega d}{h} \cdot ld \cdot d \quad (9)$$

That is:

$$M_r \sim \mu \frac{\omega d^3}{h} l \quad (10)$$

Since the distance d is eliminated on both sides, we find the same Equation after the combination of the above equation:

$$\omega \sim \left(\frac{\mu_v k \Delta T d}{\rho_v l L} \right) (\rho)^{3/4} (g\gamma)^{1/4} (\mu)^{-2} R^{-1} \quad (11)$$

Bring the typical values of the parameters into the above equation, we find that the coefficient of the equation is at the scale of 10^0 , while the coefficient deduced from the data is at the scale of 10^4 . This deviation is because we omitted a large number of constants during the calculation of the scaling law. Although a large deviation exists, the experimental data demonstrates that ω increases as R^{-1} increases, and its dependence are found to be in good agreement with equation (11), which suggests our model captures the key physical reasons for the observed rotation behaviour. (*Phys. Fluids*. 2003, 15, 1632; *Nat Phys*. 2011, 7, 395; *Europhys. Lett*. 2011;96, 58001).

4. Fig 4i,j,k: The fishbone pattern is extremely reminiscent of that proposed by Soto et al. (PRF, 2016), which, curiously, is not cited. I think it should, because the principles described in Soto's paper are close to the ones exploited by the authors. In addition, the authors should compare the dynamics they obtain to the one reported by Soto.

Response: Thank the reviewer for the comments. In the research by Soto et al., they placed a drop on a solid surface with fishbone-shaped grooves, and when the surface heated above the Leidenfrost temperature, the drop moved in the direction of the fishbone opening due to the viscous drag force generated by the vapor flow. In their research, the drop levitates on the vapour and demonstrates a complete Leidenfrost state. In our work, we utilized the patterned wettability to achieve a similar directional movement, while the drop is partly levitated and consequently displays a "compartmentalized Leidenfrost state". The moving speed of drop on the patterned wettability surface (about 4 cm s^{-1}) is smaller than the drop on physical structure surface, which may be due to the adhesion of the superhydrophilic pattern.

We have added the related discussion “As shown in Fig. 4i-j, when placed on the substrate with a “fishbone” wettability pattern, a drop moves along the pattern with a speed of 4 cm s^{-1} (Supplementary Movie 10).” on Page 10, as well as Ref. 52.

At least as surprising, the work by the authors published in Nature Communications (2019) is not cited either although it deals with very similar ideas, that is, conversion of energy into rotation using chiral patterns. Here again, it must be cited, for instance in the introduction, or when describing the rotation of the drop.

Response: Thanks for the reviewer's comments. In our previous work, the rotation was realized by impacting a drop on a patterned wettability solid surface, we think the mechanism of rotation is quite different from the current work, so we did not cite it before.

In the revised manuscript, we have added the reference (Ref. 45).

Minor details:

1. Sentences that exaggerate the applications of an effect always weaken a paper, and we have a few examples of this kind in the paper. For instance, I would suppress the sentence “and demonstrate the potential of the droplet steam engine...” in the abstract. These drops are very, very poor engines when you compare the injected energy and the resulting kinetic energy...

Response: We are very grateful to the reviewers for the important comments. We have removed the sentence about the application “and demonstrate the potential of the droplet steam engine and micro-mechanical actuator” in the Abstract.

2. Rank the ref. 5 to 7 in the chronological order. It is important to credit who had the ideas first. Same thing for ref. 11 to 14.

Response: Thanks for the reviewer's comment. We have reranked Ref. 5 to 7 and Ref. 11 to 14 in the revised manuscript.

3. Line 21 on page 5: is it 4π in the formula or 2π ?

Response: Thanks for the reviewer's comment. We have corrected this typo.

4. Line 9 on page 10: a verb is missing. Reword also the next sentence.

Response: Thanks for the reviewer's comment. We have added the verb “is”.

5. Line 12 on page 11: dargs should be corrected.

Response: Thanks for the reviewer's comment. We have corrected this typo.

Finally, we want to thank the reviewer again for these thoughtful comments. The manuscript has greatly benefited from these insightful suggestions.

Reviewer #2 (Remarks to the Author):

Language is impenetrable, with inappropriate synonyms used instead of more appropriate words. The abstract needs a thorough re-work and language check. Examples of unusual wording below, however the language needs revising throughout:

Response: We are deeply thankful to the reviewer for the time and effort in reading our manuscript and providing feedback. A throughout revision have been performed to resolve this issue:

- (1) The Abstract has been revised and the inappropriate words in the manuscript have been replaced.
- (2) The format of all equations in the manuscript has been modified.
- (3) A discussion about the comparison with existing literature has been added.
- (4) The related references have been added.

- Page 1 Line 15-16: “The vapour escaping from the Leidenfrost film causes riotous flows, and moves the droplet to wander around”

Response: The sentence was modified to “The vapour escaping from the Leidenfrost film causes uncontrollable flows, and actuates the drop to move around”.

- Line 21: “the phase-change vapour dynamics remains fragmentary”

Response: The word “fragmentary” was replaced by “incomplete”.

- Page 3 Line 7: “tranquilized boiling”

Response: The word “tranquilized” was replaced by “suppressed”.

- Line 12: “explosive boiling”

Response: The phrase was replaced by “explosively nucleate boiling”.

- Throughout: “wandering”

Response: The word was replaced by “traveling”.

- Page 5 Line 6: “ramble”

Response: The word was replaced by “move”.

- Page 12: Lines 1-2: “The droplet gyro originates from the force couple induced by symmetry-breaking vapour flows that is maneuvered through tailoring the Leidenfrost film”.

Response: The sentence was modified to “The drop gyro originates from the symmetry-breaking vapour flows that is realized by tailoring the Leidenfrost film”.

Equations need to be separated on placed on dedicated lines, rather than inline for a number of defined parameters, for example:

- Page 10 Line 5: separate the equation as parameter definition is unclear

Response: Thanks to the reviewers for the valuable comment. We have thoroughly checked the manuscript and separated all equations on dedicated lines (Page 7).

Extensive works by McHale et al with multiple institutions (Nottingham Trent, Northumbria,

Durham and Edinburgh) in the areas of the Leidenfrost propulsion and control (selective and bulk), sublimation heat engine, hydrophobicity, coatings and varying surface wettability (G-SLIPS) have not been referred to at all, which is a glaring omission which must be rectified. The literature would not be complete without this information being included.

Response: We appreciate the reviewers for the valuable comments. We have reviewed the innovative works of heat engines manufacture using the Leidenfrost effect.

Wells et al. placed a dry-ice disc on a hot turbine with a temperature above 350 °C, the dry-ice disc levitated on the solid surface, and the vapour generated by sublimation rectified by the turbine, rotating the disc and generating mechanical work (*Nature Communications*, 2015, 6, 6390).

Agrawal et al. designed a continuously operating Leidenfrost rotor by pumping water on an asymmetrically textured substrate at about 220 °C (*Applied Energy*, 2019, 240, 399-408). Afterward, they presented a thin-film boiling engine with a similar configuration that works at 250 °C, and investigated the effect of vapour pressure on the power output of the Leidenfrost heat engine. They also demonstrated the actuation at a relatively low temperature of about 200 °C by using the “cold” Leidenfrost effect (*Applied Energy*, 2021, 287, 116556).

Xu et al. demonstrated a self-propelled Leidenfrost rotor by placing a wet paper with asymmetric mass distribution on a sufficiently hot surface at 240 °C (*Applied Physics Letters*, 2019, 114, 113703).

However, in these studies, the drops or dry ice usually need to completely levitate on the substrate, making them typically work at higher temperatures (above 220 °C). Whereas in this work, the drop rotates in a “compartmentalized Leidenfrost effect”, where the drop only partially levitates, contributing to a lower operating temperature (120 °C). In addition, the strategy uses a flat chemical coating that is easier to be prepared on thin and flexible films or fragile surfaces. As shown in Fig. R3a, substrates with wettability patterns were prepared by a spraying method. Fig. R3b-d demonstrated the drops placed on the glass, metal foil, and plastic film with wettability patterns.

Fig. R3. a, Preparation of the wettability pattern by a spraying method. b-d, Demonstrations of drops placed on the glass, metal foil, and plastic film with wettability patterns that are prepared by the spraying method. Scale bars in b and c are 5 mm, and in d is 10 mm.

We have added the sentence “Extensive studies have been carried out to manufacture heat engines using the “complete” Leidenfrost effect, in which the vapour is controlled by the physical structure.” and the Ref. 47-51 on Page 9.

The sentence “Furthermore, the wettability patterns can be prepared by a spraying method, which is easier to be prepared on thin and flexible films or fragile surfaces (Supplementary Fig. 8).” is added on Page 10, and Fig. R3 was added to the Revised Supplementary materials.

The paper has some merit but requires a full review and comparison with existing literature and the language used makes it difficult to read.

Response: Thanks to the reviewers for the valuable comment. We have revised the manuscript to modify the language, and added discussions to review and compare with previous literatures.

Table R1. Comparison of the existing literature with this work.

Literature	Mechanism	Rotate speed	Temperature	Reference
Phys. Fluids 2013, 25, 051704.	Leidenfrost effect	46 rpm	320 °C	47
Nat. Commun. 2015, 6, 6390.	Leidenfrost effect	60 rpm	500 °C	48
Appl. Energ. 2019, 240, 399	Leidenfrost effect	94 rpm	>220 °C	49
Appl. Phys. Lett. 2019, 114, 113703.	Leidenfrost effect	143 rpm	420 °C	50
Appl. Energ. 2021, 287, 116556.	Leidenfrost effect	162 rpm	325 °C	51
Appl. Energ. 2021, 287, 116556.	“Cold” Leidenfrost effect	38 rpm	200 °C	51
This work	Compartmentalized Leidenfrost effect	1400 rpm	120 °C	×

Finally, we thank the reviewer for these thoughtful and insightful comments. The manuscript has greatly benefited from these comments.

Reviewers' comments:

Reviewer #1 (Remarks to the Author):

I carefully read the new version of the manuscript, and I am sorry to conclude that the present state of the work does not (yet?) warrant a publication in Nature Communications, in my opinion. I still consider the results as new and interesting, I thank the authors for having taken into account many of my remarks, but the new version introduced new problems, which I now would like to discuss.

Major scientific issue

The authors tried to build a model (pages 6 and 7), which is commendable, but the model is hard to follow and not really convincing. Let us start by the major problem. When we build a model, we test it by looking at the influence of one or several parameters, which the authors did by varying the drop size and the pattern width. They claim that there is a good agreement (figure 2e), but the drop radius was varied by roughly 25%, a small variation for which we would generally expect a linear variation. The dimension l was varied by a factor of 2, but it does not impact the rotation speed by the same amount. But the main problem seems to be the order of magnitude (something we must ALWAYS calculate when we test a scaling law). I found that eq 5 predicts a rotation speed of typically 1 rad/s, by far too low compared to the observations. And indeed, the model is often questionable, for instance the choice of h in eq 4, that is, a distance characterizing the vapor thickness used here for characterizing a liquid thickness. It is nice to try scaling laws, but every choice must be justified.

From these points of view, it seems very risky to claim that the model is "quantitative" (page 6) or that "it captures the essence of the drop spin dynamics" (page 8).

Marginal note

On page 3, lines 13 and 4: give the typical rotation speed; then calculate the Weber number, $\rho\omega^2 R^3 / \sigma$ (where σ is the surface tension), which I find to be logically of order 1, in qualitative agreement with the conclusion in line 14.

Style

The paper should be carefully rewritten. Some sentences have no verb, for instance (for instance ...since the weak vaporization, on page 8). And wording is often imprecise. Here a few examples on pages 2/3 : performance -> behavior; suppress chaotic (the vapor flow is laminar); suppress a new discovery; a constructed -> the. Give the drop volume in the caption of fig 1b/d, and more generally, edit all captions with appropriate information. Just to give one example, we don't know how the figure 2e is obtained: is it from figure 2b, where the drop should shrink as time goes on? On page 6, in -> from, contact on -> contacts, define l when it appears. Very strange title (Discussion) on page 11 for a paragraph which is not at all a discussion.

Reviewer #2 (Remarks to the Author):

Thank you for the updated manuscript, which is much improved.

A few further comments, one of the corrections by the other reviewer mentions "fishbone" which has been added to the manuscript. This should be "herringbone" throughout, I believe this may have been a translation issue. Also, please see work by Dodd et al. (part of the McHale group research, "Planar Selective Leidenfrost Propulsion Without Physically Structured Substrates or Walls", and "Low friction self-centering droplet propulsion and transport using a Leidenfrost herringbone-ratchet structure"), who have extensive work both on herringbone substrates and planar Leidenfrost substrates. References to these works should be included to provide a comparison on the progress in using the Leidenfrost effect in non-standard ways to improve usefulness of the effect.

Upon completion of this minor corrections, I am happy for this to be published.

Reviewer #1 (Remarks to the Author):

Comment #1: I carefully read the new version of the manuscript, and I am sorry to conclude that the present state of the work does not (yet?) warrant a publication in Nature Communications, in my opinion. I still consider the results as new and interesting, I thank the authors for having taken into account many of my remarks, but the new version introduced new problems, which I now would like to discuss.

Response: We deeply thank the reviewer for the time and effort in thoroughly reading our manuscript again. Especially, we are very appreciative of the reviewer's interest in our work, as well as providing constructive suggestions, which will definitely improve our work. Encouraged and motivated by these comments, we have carefully revised our model, and made necessary modifications to our manuscript. We believe that the resubmitted manuscript would dispel the concerns of the reviewer.

Comment #2: The authors tried to build a model (pages 6 and 7), which is commendable, but the model is hard to follow and not really convincing. Let us start by the major problem. When we build a model, we test it by looking at the influence of one or several parameters, which the authors did by varying the drop size and the pattern width. They claim that there is a good agreement (figure 2e), but the drop radius was varied by roughly 25%, a small variation for which we would generally expect a linear variation. The dimension l was varied by a factor of 2, but it does not impact the rotation speed by the same amount. But the main problem seems to be the order of magnitude (something we must ALWAYS calculate when we test a scaling law). I found that eq 5 predicts a rotation speed of typically 1 rad/s, by far too low compared to the observations. And indeed, the model is often questionable, for instance the choice of h in eq 4, that is, a distance characterizing the vapor thickness used here for characterizing a liquid thickness. It is nice to try scaling laws, but every choice must be justified.

From these points of view, it seems very risky to claim that the model is "quantitative" (page 6) or that "it captures the essence of the drop spin dynamics" (page 8).

Response: Thanks for the insightful comments. We have seriously considered and adopted these suggestions, as elaborated below in detail.

Firstly, as suggested by the reviewer, we have added experimental results to verify the correctness of our model. We measured the mass change of the drop after placing it on the hot substrate with a wettability pattern, as shown in Fig. R1. The drop evaporation rate is about 1.2 mg s^{-1} during the drop rotation process (29 s to 112 s).

Figure R1. The mass change of the drop. The orange area indicates the period of drop rotation.

In our last model, the drop evaporation rate was calculated by:

$$\dot{m} = \frac{k\Delta Tld}{L\delta_t} \quad (1)$$

where k , L , ΔT , R , d , and l are the thermal conductivity, the vaporization latent heat of water, the temperature difference, the drop radius, the pattern side length, and the width of the pattern. And δ_t is the thickness of the thermal boundary layer, that is $\delta_t \approx \delta \sim \sqrt{l\mu/\omega\rho R}$. We take typical values $\omega = 80 \text{ rad s}^{-1}$ and $R = 2 \text{ mm}$, and obtain the drop evaporation rate $\dot{m}(\text{cal}) \approx 0.1 \text{ mg s}^{-1}$, which is much smaller than the experimental value of 1.2 mg s^{-1} . The difference between the experimental and the calculated values originated from our assumption of steady-state heat transfer with no boiling, whereas in reality a violent contact boiling occurs at the superhydrophilic region under the drop. This underestimated evaporation rate leads to a slow vapor flow at the bottom of the drop and generates a weak driving force, which causes the calculated rotational speed (1 rad s^{-1}) to be much smaller than the experimental result (100 rad s^{-1}). We are very grateful to the reviewer for pointing out the problem that drives us to reconsider our model.

Initially, we calculate the heat transfer between the substrate and the drop to get the vapor flow speed under the drop (U), but the existence of contact boiling significantly complicates this process. According to the reviewer's comment in the 1st round of the Review Report: " U would just be a mean velocity of vapor produced by the line, and not explicitly given by an equation", we try to combine the experimental results with the theoretical calculations. As shown in Fig. R1, the evaporation rate of the drop is measured to be roughly constant during the rotation process, which allows us to use the measured evaporation rate \dot{m} in the model to calculate the drop rotational speed. The updated model is detailed as follows.

Assuming the vapor is incompressible in our system, the vapor flow speed U under the drop can be obtained by:

$$\dot{m} \approx 2\pi R U h \rho_v \quad (2)$$

where \dot{m} is the measured drop evaporation rate, h is the thickness of the vapor layer, ρ_v is the density of the vapor.

The lubrication approximation equation links the pressure ΔP to U and h , that is:

$$\frac{\Delta P}{R} \approx 12\mu_v \frac{U}{h^2} \quad (3)$$

where μ_v is the viscosity of the vapor. For the drop with a radius of 2 mm, ΔP can be considered as the hydrostatic pressure under the drop, thus $\Delta P \approx 2\rho g l_c$. At 100°C , the capillary length of water l_c is about 2.5 mm, resulting in a pressure of 50 Pa under the drop.

Combining the above equations, we obtain U and h :

$$U = \frac{(\rho g l_c)^{1/3} \dot{m}^{2/3}}{2\pi R \mu_v^{1/3} \rho_v^{2/3}} \quad (4)$$

$$h = \left(\frac{\mu_v \dot{m}}{\rho g \rho_v l_c} \right)^{1/3} \quad (5)$$

Typical values for the parameters are: $\rho \approx 10^3 \text{ kg m}^{-3}$, $g \approx 9.8 \text{ m s}^{-2}$, $l_c \approx 2.5 \times 10^{-3} \text{ m}$, $\dot{m} \approx 1.2 \times 10^{-6} \text{ kg s}^{-1}$, $R \approx 2 \times 10^{-3} \text{ m}$, $\mu_v \approx 10^{-5} \text{ Pa s}$, and $\rho_v \approx 1 \text{ kg m}^{-3}$. Hence, we find $h = 8 \times 10^{-5} \text{ m}$, and $U = 1.2 \text{ m s}^{-1}$.

Then, we can calculate the driving force for the rotation when considering the segmented Leidenfrost films as a rectangle, that is:

$$F_d \approx \mu_v \frac{U}{h} \times 2d^2 = \frac{(\mu_v \dot{m})^{1/3} (\rho g l_c)^{2/3}}{\pi \rho_v^{1/3} R} d^2 \quad (6)$$

As shown in Fig. R2, we can simplify the calculation by assuming that the driving force acts at the corners of the pattern. Hence, the driving torque M_d is:

$$M_d \approx 2 \sin \pi/4 \cdot d F_d \quad (7)$$

We find $F_d \approx 7 \times 10^{-7}$ N, and $M_d \approx 1.5 \times 10^{-9}$ Nm with $d = 1.5$ mm. The results are comparable with the previous studies of Leidenfrost drops driven on ratchet surfaces (*Nat. Phys.* 2011, 7, 395).

Figure R2. Scheme of driving forces of the Leidenfrost drop on the patterned surface.

Based on the reviewer's comments, we have carefully examined the resisting force. In our system, the drop levitates in the superhydrophobic region, and contacts the superhydrophilic pattern. Therefore, mainly two velocity gradients exist in the system that can generate viscous resisting force: the velocity gradient in the drop (Fig. R3a) and the velocity gradient in the liquid bridge (Fig. R3b). The resisting torque M_r is:

$$M_r = M_{r1} + M_{r2} \quad (8)$$

where M_{r1} is the resisting torque inside the drop and M_{r2} is the resisting torque inside the liquid bridge.

Figure R3. Scheme of velocity gradients existing in the system. (a) In the drop. (b) In the liquid bridge.

For the resisting torque generated by the velocity gradient in the drop:

We have placed plastic particles with a radius of $70 \mu\text{m}$ on the surface of a rotating drop to roughly estimate the velocity gradient in the direction of drop thickness, as shown in Fig. R4. We find that the particles on the top surface rotate $3/4$ round when the drop makes a round, indicating a velocity gradient of $\omega/8l_c$ exists in the drop thickness. This gives the resisting torque generated inside the drop is:

$$M_{r1} \approx \int_0^R \mu \frac{\pi \omega}{4l_c} r^3 dr = \frac{\pi \mu \omega R^4}{16l_c} \quad (9)$$

We find M_{r1} is about 10^{-10} Nm with $\omega = 80 \text{ rad s}^{-1}$, $R = 2 \text{ mm}$, which is much smaller than the driving torque.

Figure R4. Velocity gradient exiting in the drop.

For the resisting torque generated by the velocity gradient in the liquid bridge:

Since the liquid bridge connects the rotating drop and the static substrate, a velocity gradient from 0 to ω exists over the thickness of the bridge, which equals the thickness of the vapor layer h . To simplify the calculation, we divide the superhydrophilic pattern into two parts (I and II as shown in Fig. R5a), and consider II as a segment of an arc (Fig. R5b). Hence, the resisting torque M_{r2} is:

$$M_{r2} \approx 2 \int_0^d \mu \frac{\omega r}{h} l \cdot r dr + 2\mu \frac{\omega l d^3}{h} = \frac{8 \mu \omega l d^3 (\rho g l_c)^{1/3}}{3 (\mu_v \dot{m})^{1/3}} \quad (9)$$

We find this resisting torque is about 1.1×10^{-9} Nm, which is comparable with the driving force. Therefore, we can consider that:

$$M_r = M_{r1} + M_{r2} \approx \frac{8 \mu \omega l d^3 (\rho g l_c)^{1/3}}{3 (\mu_v \dot{m})^{1/3}} \quad (10)$$

Combining equations (6), (7), and (8) gives the rotating speed ω is:

$$\omega \sim a l^{-1} R^{-1} \quad (11)$$

where $a = (\mu_v \dot{m})^{2/3} (\rho g l_c)^{1/3} / 6 \mu \rho_v^{1/3}$. Typical values for the parameters are: $\mu_v \approx 10^{-5}$ Pa s, $\dot{m} \approx 1.2 \times 10^{-6}$ kg s $^{-1}$, $\rho \approx 10^3$ kg m $^{-3}$, $g \approx 9.8$ m s $^{-2}$, $l_c \approx 2.5 \times 10^{-3}$, $\rho_v \approx 1$ kg m $^{-3}$, $l \approx 10^{-4}$ m, and $R \approx 2 \times 10^{-3}$ m. We find $a \approx 2.5 \times 10^{-5}$ m 2 , and ω is at the scale of 100 rad s $^{-1}$, which coincides with the experimental data.

Figure R5. Division of the superhydrophilic pattern.

Further, we demonstrate the validity of the model with additional experiments. According to the reviewer's comment, we try to increase the range of the drop radius variation as large as possible. However, when R is increased to larger than 2.6 mm, the large inertial force deforms the drop irregularly which seriously affects the rotation, while when R is decreased to smaller than d , the drop stops rotating. As shown in Fig. R6, the droplet radius R varies from 1.5 mm to 2.6 mm (variation by 42%), which is larger than that of 1.5 mm to 2 mm in the previous version of the manuscript (variation by 25%). The rotational speed ω shown in Fig. R6 demonstrates a linear decrease with the increase of R^{-1} , which is consistent with the updated model. Moreover, ω decreases as l increases. Taking the drop with a radius of 2 mm as an example. ω decreases from 90 rad s $^{-1}$ to 58 rad s $^{-1}$ when l changes from 70 to 130 μ m, which is slightly deviated from the theoretical result. This

may be because \dot{m} also enlarges as l increases. We have revised the model in the manuscript and the supplementary material.

Figure R6. The variation of ω versus R^{-1} .

Modifications in the manuscript:

1. On page 6, line 13: we have deleted the word “quantitatively” in the sentence “To confirm the hypothesis, we build a model to understand this phenomenon.”
2. On pages 6 to 8: we have revised the theoretical model accordingly.
3. On page 8, line 15, we have deleted the sentence “the well-coincidence between the experimental data (dots) and theoretical results (dashed lines) indicates that the model captures the essence of the drop spin dynamics.”
4. On page 22, we have updated Fig. 2d to indicate the liquid bridge.
5. On page 22, we have added the curve of drop mass versus time in Fig. 2e.
6. On page 22, we have updated Fig. 2f with an enlarged variation range of R .

Comment #3: On page 3, lines 13 and 4: give the typical rotation speed; then calculate the Weber number, $\rho\omega^2 R^3/\sigma$ (where σ is the surface tension), which I find to be logically of order 1, in qualitative agreement with the conclusion in line 14.

Response: We thank the reviewers for confirming our conclusion through the calculation of the Weber number. We find that $We = 1.1$ with $R = 2$ mm and $\omega = 100$ rad s^{-1} , which means the centrifugal force is comparable with the surface tension. We have modified the related discussion in the manuscript.

Modification to the manuscript:

On page 3, lines 12-16, we have modified the discussion to “As the snapshots of the synchronized oblique and side views displayed in Fig. 1d, spontaneous drop spin with a speed of 800 rpm occurs on the chemically heterogeneous substrates. Considering the drop with a radius of 2 mm, the Weber number $We = \rho\omega^2 R^3/\sigma$ is of order 1, implying that the balance between centrifugal force and surface tension shapes the drop ellipsoidal.”

Comment 4: The paper should be carefully rewritten. Some sentences have no verb, for instance (for

instance ...since the weak vaporization, on page 8). And wording is often imprecise. Here a few examples on pages 2/3 : performance -> behavior; suppress chaotic (the vapor flow is laminar); suppress a new discovery; a constructed -> the.

Response: Thanks for the valuable comment. A throughout check was performed to correct the typos and inappropriate words.

Modification to the manuscript:

- On page 1, line 23: the word “suppress” was replaced by “reduce”.
- On page 2, line 5: the word “performance” was replaced by “behaviour”.
- On page 2, line 8: the word “since” was replaced by “because of”.
- On page 2, line 8: the word “suppressed” was replaced by “slowed”.
- On page 2, line 9: the word “chaotic” was deleted.
- On page 2, line 20: the word “chaotic” was replaced by “generated”.
- On page 3, line 7: the word “a” was replaced by “the”.
- On page 8, line 20: the word “since” was replaced by “due to”.
- On page 9, line 1: the word “since” was replaced by “contribute to”.
- On page 23, line 7: the word “the” was deleted.
- On page 24, line 7: the word “since” was replaced by “due to”.
- On page 24, line 7: the typo “palced” was revised.

Give the drop volume in the caption of fig 1b/d, and more generally, edit all captions with appropriate information. Just to give one example, we don't know how the figure 2e is obtained: is it from figure 2b, where the drop should shrink as time goes on?

Response: Thanks for the valuable comment. We have carefully revised the captions and added appropriate information.

Modification to the manuscript:

- On page 21, line 1: the sentence “The data is collected from the rotating process of **d**.” was added.
- On page 21, line 2: the sentence “The volume of the drops in **b** and **d** is 60 μl .” was added.
- On page 23, lines 2-3: the sentences “When placed on the substrate, the volume of the drop is 80 μl . The temperature of the substrates is 120 $^{\circ}\text{C}$.” were added.
- On page 23, lines 7-9: the sentences “**e**, Mass change of the heated drop. The initial volume of the drop is 80 μl , and the temperature of the substrates is 120 $^{\circ}\text{C}$. The rate of mass change during the rotating process (the brown region) is about 1.2 mg s^{-1} .” were added.
- On page 23, lines 10-11: the sentence “Data for each width are collected from seven droplet rotation videos.” was added.
- On page 24, line 8: the sentence “The initial volumes of drops in **a** and **d** are 60 μl , and in **b** and **c** are 10 μl .” was added.
- On page 26, line 5: the sentence “The volume of drops in **j** and **k** is 60 μl .” was added.

On page 6, in -> from, contact on -> contacts, define *l* when it appears.

Response: Thanks for the valuable comment. We have carefully revised the grammatical errors, and added the definition of *l* when it appears.

Modification to the manuscript:

On page 6, line 7: the word “in” is revised to “from”.

On page 6, line 7: the word “contacts” is revised to “contact”.

On page 8, line 4: the sentence “where l is the width of the superhydrophilic pattern.” is added.

Very strange title (Discussion) on page 11 for a paragraph which is not at all a discussion.

Response: Thanks for the valuable comment. We have modified the title.

Modification to the manuscript:

On page 11, line 13: the title “Discussion” is revised to “Conclusion”.

Finally, we want to thank the reviewer again for these thoughtful and insightful comments. The manuscript has greatly benefited from these valuable comments.

Reviewer #2 (Remarks to the Author):

Comment 1: Thank you for the updated manuscript, which is much improved.

A few further comments, one of the corrections by the other reviewer mentions "fishbone" which has been added to the manuscript. This should be "herringbone" throughout, I believe this may have been a translation issue.

Response: We deeply appreciate the reviewer for the time and effort in reading our manuscript, as well as the positive comments. We have thoroughly checked the manuscript and revised the word “fishbone” to “herringbone”.

Modification to the manuscript:

On page 6, line 7: the word “fishbone” is modified to “herringbone”.

On page 26, line 3: the word “fishbone” is modified to “herringbone”.

Also, please see work by Dodd et al. (part of the McHale group research, "Planar Selective Leidenfrost Propulsion Without Physically Structured Substrates or Walls", and "Low friction self-centering droplet propulsion and transport using a Leidenfrost herringbone-ratchet structure"), who have extensive work both on herringbone substrates and planar Leidenfrost substrates. References to these works should be included to provide a comparison on the progress in using the Leidenfrost effect in non-standard ways to improve usefulness of the effect.

Upon completion of this minor corrections, I am happy for this to be published.

Response: Thanks for the valuable comment. We have carefully read the literature mentioned by the reviewer. In the first study, the authors propose an innovative method to levitate and actuate droplets by using planar lithography. The patterned electrodes can be heated, generating controllable pressure and vapour flows. Benefiting from the almost featureless structure, the velocity of the droplets can achieve approximately 30 mm s^{-1} . In the second study, the authors present a negative feedback strategy to solve the problems of low

friction and poor control of Leidenfrost droplets. They designed a herringbone and ratchet combined structures, which create differential forces at the bottom of off-center droplets, thus keeping the droplet moving centrally. These interesting studies will definitely make the readers understand this work more comprehensively. Therefore, we have included these references and the related discussion in the manuscript.

Modification to the manuscript:

On page 11, line 10: the sentence “As shown in Fig. 4i-j, when placed on the substrate with a “herringbone” wettability pattern, a drop moves along the pattern with a speed of 4 cm s^{-1} (Supplementary Movie 10), which is comparable with the Leidenfrost drops on micro-patterned electrodes.” is added.

On page 18, lines 16-19: References 53 and 54 were added.

Lastly, we would like to thank the reviewers again. The manuscript benefited greatly from these thoughtful and valuable comments.

REVIEWER COMMENTS

Reviewer #1 (Remarks to the Author):

I was happy of this opportunity to read again this work by Li et al. As a main result, the authors show that drops on hydrophobic surfaces can be in a mixed state (contacting and no-contacting their substrate), which they exploit to induce motion in the drop. This result is nice and new and it deserves a publication in Nature Communications. I recommend that the authors take into account the following remarks prior to publication, in case the editor agrees with my recommendation.

Line 8, page 2: Boiling often refer to bubbles – no bubbles in the Leidenfrost state. To avoid the confusion, replace by evaporation.

Line 14, page 3: Please give typical values of ω in rad/s in addition to the radius so that we can evaluate the Weber number.

Line 7, page 4: Too short. It must be emphasized that the new system only operates in a limited range of temperature that corresponds to the range between the “cold Leidenfrost state” and the “normal Leidenfrost state”, that is, in a range where the hydrophilic region is wetted by the liquid, but not the hydrophobic one. We can see this limited range as a drawback (the system works only in a small interval of temperature) or as an advantage (the Leidenfrost effect has a very low efficiency for a motor, when we see the injected energy and the resulting energy, but this case is more favorable because it works at “low” temperatures, close to the boiling point of water).

The functional domain should be given again in the conclusion, in term of temperature – if not, it seems as something very general, which it is not, since it exploits the narrow domain where hydrophilic and hydrophobic solids do not behave the same way, from the Leidenfrost point of view.

Line 7, page 6: Hydrophilic, and not hydrophilic.

Line 15, page 7: Provide the unit of l_c .

Reviewer #1 (Remarks to the Author):

I was happy of this opportunity to read again this work by Li et al. As a main result, the authors show that drops on hydrophobic surfaces can be in a mixed state (contacting and no-contacting their substrate), which they exploit to induce motion in the drop. This result is nice and new and it deserves a publication in Nature Communications. I recommend that the authors take into account the following remarks prior to publication, in case the editor agrees with my recommendation.

Response: We sincerely thank the reviewer for the time and effort in reviewing our manuscript again. Especially, we are very appreciative of the reviewer's interest in our work, as well as the positive comments.

Line 8, page 2: Boiling often refer to bubbles – no bubbles in the Leidenfrost state. To avoid the confusion, replace by evaporation.

Response: Thanks for the valuable comment. We have replaced the word “boiling” by “evaporation” in Line 8, Page 2 of the revised manuscript.

Line 14, page 3: Please give typical values of omega in rad/s in addition to the radius so that we can evaluate the Weber number.

Response: Thanks for the valuable comment. We have provided the rotational speed of 84 rad/s in Line 14, Page 3 of the revised manuscript.

Line 7, page 4: Too short. It must be emphasized that the new system only operates in a limited range of temperature that corresponds to the range between the “cold Leidenfrost state” and the “normal Leidenfrost state”, that is, in a range where the hydrophilic region is wetted by the liquid, but not the hydrophobic one. We can see this limited range as a drawback (the system works only in a small interval of temperature) or as an advantage (the Leidenfrost effect has a very low efficiency for a motor, when we see the injected energy and the resulting energy, but this case is more favorable because it works at “low” temperatures, close to the boiling point of water).

The functional domain should be given again in the conclusion, in term of temperature – if not, it seems as something very general, which it is not, since it exploits the narrow domain where hydrophilic and hydrophobic solids do not behave the same way, from the Leidenfrost point of view.

Response: Thanks for the valuable comment. Accordingly, we have added discussions to emphasize that the new system only operates in a limited range of temperature, as elaborated below.

Line 7, Page 4: “Fig. 1f summarizes the substrate temperature and the drop volume required for the drop spin. The droplet spinning system operates in a limited temperature range from 115 °C to 135 °C, which is between the “cold Leidenfrost state” and the “normal Leidenfrost state”, that is, in a range where the hydrophilic region is wetted by the liquid, but not the hydrophobic one.” were added.

Line 17, Page 11: “Benefiting from the “compartmentalized Leidenfrost state” achieved by the wettability pattern, this strategy is effective in a specific “low” operating temperature interval, which implies that it may have a higher energy conversion efficiency compared to the “normal Leidenfrost effect”.” were added.

Line 7, page 6: Hydrophilic, and not hydriphilic.

Response: Thanks for the valuable comment. We have corrected the typo in Line 11, Page 6 of the revised manuscript.

Line 15, page 7: Provide the unit of l_c .

Response: Thanks for the valuable comment. We have added the unit “m” of l_c in Line 20, Page 7 of the revised manuscript.